# Which Measuring Method Is Better for Reflecting Subtalar Joint Stiffness?

**DOI:** 10.3390/jcm14061887

**Published:** 2025-03-11

**Authors:** You-Keun Kim, Myoungyeol Shin, Jaehyeon Seo, Sun-Kyoung Lee, Ho-Seong Lee

**Affiliations:** 1Department of Orthopedic Surgery, Bupyeong Himchan Hospital, Incheon 21399, Republic of Korea; rladbrms2@naver.com; 2Department of Orthropedic Surgery, National Police Hospital, Seoul 05715, Republic of Korea; trsinml@nate.com; 3Department of Orthopedic Surgery, Asan Medical Center, University of Ulsan College of Medicine, 88, Olympic-ro 43-gil, Songpa-gu, Seoul 05505, Republic of Korea; jhseo112@gmail.com; 4Department of Orthopedic Surgery, Chungbuk National University Hospital, Cheongju-si 28644, Republic of Korea; sunkl0715@gmail.com

**Keywords:** range of motion, stiffness, subtalar joint

## Abstract

**Background/Objectives**: A standard method for the measurement of subtalar joint motion has yet to be developed. This study aimed to determine which of the two methods (AMA and modified McBride) for the measurement of subtalar joint motion better reflects subtalar joint stiffness. **Methods**: We evaluated the intraobserver and interobserver reliability and validity of the two methods in the normal group (*n* = 50) and the patient group (*n* = 90; 30 with normal hindfoot, 30 with ankle joint pathology, and 30 with subtalar joint pathology). We assessed the intraclass correlation coefficient (ICC) and receiver operating characteristic (ROC) curve for statistical analysis. **Results**: In the normal group, the intraobserver reproducibility and interobserver reliability of the AMA method were better than those of the modified McBride method. In the hindfoot patient group, both methods showed excellent interobserver reliability (ICC > 0.75). The results of the ROC curve also indicated that the modified McBride method (AUC: 0.919, 0.813) was better than the AMA method (AUC: 0.715, 0.749) in reflecting subtalar joint stiffness. **Conclusions**: In patients with subtalar joint lesions, the modified McBride method had a relatively better validity and was more appropriate than the AMA method for measuring the stiffness of the subtalar joint.

## 1. Introduction

Osteoarthritis (OA) is one of the most common causes of disability around the globe, especially in aging populations. The main symptoms of OA are pain and the loss of motion and function of the affected joint [1]. The range of motion (ROM) and stiffness are basic physical findings that reflect joint pathology. Therefore, the consistent measurement of the ROM is mandatory for evaluating a patient’s problem and providing academic communication in research. However, unlike measurement methods for other joints, standardized methods have yet to be developed for determining the ROM that reflects the pathology of the subtalar joint; furthermore, a clear reference value for the normal ROM of the subtalar joint is not yet available [2,3]. These discrepancies are attributed to the following: (1) the lack of an external landmark of the subtalar joint, (2) the three-dimensional orientation of the subtalar joint motion axis, and (3) compositive motion with the adjacent joint, including the ankle joint and Chopart’s joint.

Previous studies demonstrated the measurement of the subtalar joint ROM. For instance, McMaster measured the ROM of the subtalar joint in a supine position by attaching a pointer to the heel and drawing lines on a blackboard at the limits of subtalar inversion and eversion. The normal ranges for subtalar inversion and eversion are 25° and 5°, respectively [4]. Elveru measured the ROM of the subtalar joint in a prone position with a goniometer attached to the heel. They also determined the subtalar motion during the inversion and eversion of the heel [5]. However, these measurement methods and normal ranges vary across studies.

Subtalar joint motion can be measured in front of the patient with knee flexion (McBride method) or the posterior side of the patient’s heel in a prone position (AMA method). The AMA and McBride methods are commonly used to measure the subtalar joint ROM in outpatient settings. However, studies have yet to determine which of these two methods better reflect subtalar joint stiffness. Therefore, this study aimed to compare two commonly used physical examination methods (AMA and modified McBride) for measuring the subtalar joint ROM and determine the superiority of either examination in the evaluation of the subtalar joint pathology, including stiffness. This study has the following hypothesis: the modified McBride method reflects the stiffness of the subtalar joint better than the AMA method, which will be verified by ICC and AUC analysis.

## 2. Materials and Methods

In this study, the reliability of the measurement method in normal young people was initially evaluated, and the reliability and validity of the patient group were subsequently examined.

### 2.1. Measurement Method

The subtalar joint motion was measured using two methods as described below.

(1) AMA method:

In accordance with the American Medical Association (AMA), subtalar joint motion was measured with the knee flexed to 90° in a prone position [6]. The ankle joint motion was restricted by a slightly dorsiflexed ankle joint, and the subtalar joint was inverted or everted. The angle was measured in the posterior aspect between the central line of the lower leg and the central line of the hindfoot while inverting or everting. This angle was defined as the inversion or eversion angle (Figure 1).

Subtalar joint motion was measured with the knee flexed to a 90-degree angle in a prone position. The degree of inversion and eversion was measured using the central line of the lower leg and the hindfoot.

(2) Modified McBride method:

In the original McBride method, the degree of the plantarflexion of the ankle joint is not standardized [7]. Therefore, in this study, it was modified (modified McBride method).

Subtalar joint motion was measured in a sitting position, with the plantar surface in contact with the ground, the knee flexed to 90°, and the foot placed on a 3 cm high pad on the heel, thereby allowing the ankle joint to be plantarflexed to some extent. The connecting line in the ankle center and the second toe was used to measure the degree of inversion and eversion (Figure 2).

Subtalar joint motion was measured in a sitting position where the plantar surface was in contact with the ground with the knee flexed to a 90-degree angle. With the foot placed on a 3 cm high pad on the heel, the ankle joint was plantarflexed to some extent. The line connecting the ankle center (Black line) and the 2nd toe (yellow line) was used to measure the degree of inversion and eversion.

### 2.2. Measurement Subjects

(1) Normal group

The subjects were 50 young volunteers with a mean age of 23.2 years (range, 20–26 years) and without any acute or chronic lower leg pain. Exclusion criteria were as follows: generalized laxity, flat foot, and chronic ankle instability. The presence of generalized laxity was assessed using the Beighton 9-point scoring system for all [8]. A flat foot is characterized by a clear medial arch loss on physical examination or a talo-first metatarsal angle, which is also known as a Meary’s angle of greater than 4 degrees with the apex oriented inferiorly in a standing foot lateral radiographic [9]. Chronic ankle instability was defined as the history of recurrent sprain, more than grade II instability on physical examination, and definite instability on a stress ankle radiograph [10]. Subtalar joint motion was independently measured by two examiners (examiner I: foot–ankle specialist; examiner II: well-trained clinical nurse) using two methods. Measurements were performed twice with a minimum interval of 2 weeks. After the measurement, the intraobserver reproducibility and interobserver reliability were assessed.

(2) Patient group

Subtalar joint motion was independently measured by two different foot–ankle specialists in 90 patients representing the patient group at a foot–ankle clinic. This group was categorized into three subgroups based on radiographic findings: (1) NL group, 30 patients with a normal ankle–subtalar joint complex; (2) AN group, 30 patients with localized ankle joint pathology (21 patients had an isolated fused ankle, and 9 patients were diagnosed with advanced ankle joint osteoarthritis with stiffness); and (3) ST group, 30 patients with localized subtalar joint pathology (5 patients had an isolated fused subtalar joint, and 25 patients had post-traumatic subtalar arthritis after calcaneal fracture). Demographic characteristics in three groups are shown in Table 1.

### 2.3. Statistical Analysis

First, intraobserver reproducibility and interobserver reliability were determined using the intraclass correlation coefficient (ICC) test with absolute agreement. The ICC was interpreted according to the following criteria of Fleiss [11]: excellent (ICC ≥ 0.75), fair-to-good (0.40 ≤ ICC < 0.75), and poor (ICC < 0.40). Second, the average measured values of the three subgroups were compared using the Mann–Whitney test. Statistical significance was considered when the *p*-value was <0.05. Third, the area under the curve (AUC) on the receiver operating characteristic (ROC) curve was calculated. The ROC curve refers to the plot of the true positive rate against the false positive rate for the different possible cutpoints of the two methods. A higher ROC curve indicates a better reliability. The AUC was interpreted on the basis of the following criteria of Swets [12]: excellent (AUC ≥ 0.90), good (0.80 ≤ AUC < 0.90), fair (0.70 ≤ AUC < 0.80), poor (0.60 ≤ AUC < 0.70), and fail (AUC < 0.60). Data were statistically analyzed using IBM SPSS version 22.0 (IBM Corp., Armonk, NY, USA), and data with *p* < 0.05 were considered significant.

## 3. Results

### 3.1. Normal Group

(1) Mean ROM

The mean ROM measured by examiner I was as follows. With the AMA method, the mean inversion and eversion angles were 18.3° (15–21°) and 9.8° (7–12°), respectively. With the modified McBride method, the mean inversion and eversion angles were 29.3° (25–33°) and 11.8° (9–13°), respectively.

(2) Intraobserver reproducibility—ICC (Table 2)

With examiner I, the AMA method showed an excellent result, and the modified McBride method showed a fair-to-good result. With examiner II, both methods exhibited fair-to-good results. The intraobserver reproducibility of examiner I was better than that of examiner II. Furthermore, the reproducibility of the AMA method was better than that of the modified McBride method.

(3) Interobserver reliability—ICC (Table 3)

Both methods showed excellent interobserver reliability between the two examiners (examiner I: foot–ankle specialist; examiner II: well-trained clinical nurse) in the normal group. In the measurement, the interobserver reliability of the AMA method was better than that of the modified McBride method.

### 3.2. Patient Group

(1) Mean ROM (Table 4)

The mean value measured by the two examiners was as follows. In the NL group, the mean inversion/eversion angles obtained by the AMA and modified McBride methods were 18.4°/9.9° and 29.2°/11.9°, respectively. In the AN group, the mean inversion/eversion angles obtained by the AMA and modified McBride methods were 9.7°/6.4° and 20.3°/10.9°, respectively. In the ST group, the mean inversion/eversion angles obtained by the AMA and modified McBride methods were 10.1°/5.7° and 15.0°/8.4°, respectively.

(2) Interobserver reliability—ICC (Table 5)

Both methods showed excellent interobserver reliability between the two examiners (foot–ankle specialists) in the patient group.

(3) Comparison of validity among three groups (Table 6)

The ROM of the AN and ST groups significantly decreased compared with that of the NL group with the two methods (*p* < 0.001). The comparison of the AN and ST groups revealed a significant difference with the modified McBride method (*p* < 0.001) but not with the AMA method (*p* > 0.05). This finding indicated that the difference between the stiff ankle joint alone and stiff subtalar joint alone was not significant in the AMA method. Therefore, the modified McBride method could reflect the subtalar joint stiffness more efficiently than the AMA method.

(4) ROC curve

The results of the ROC curve are shown in Figure 3.

The inversion angle of the modified McBride method showed an excellent result in the AUC of 91.9% (95% CI: 86.4–97.4%). The eversion angle of the modified McBride method showed a good result in the AUC of 81.3% (95% CI: 71.8–90.9%). The inversion and eversion angles of the AMA method showed fair results in the AUCs of 71.5% (95% CI: 61.0–81.9%) and 74.9% (95% CI: 64.4–85.3%), respectively. The results of the ROC curve revealed that the modified McBride method showed excellent results based on the AUC and 95% confidence interval. Furthermore, the modified McBride method was better than the AMA method in reflecting subtalar joint stiffness.

## 4. Discussion

The subtalar joint is a complex joint with multiplane motion. Therefore, subtalar joint ROM should be accurately measured. A preliminary study described the biomechanics and kinematics of the ankle joint, including the subtalar joint in in vivo, in vitro, and cadaveric studies. Allinger and Engsberg measured subtalar joint motion by using a specially designed device and obtained mean inversion and eversion angles of 18° and 11°, respectively [13]. Wang et al. measured subtalar joint motion and obtained mean inversion and eversion angles of 25–30° and 5–10°, respectively [14]. Various results on the ROM of the subtalar joint were previously reported. Recently, four-dimensional computed tomography (4-D CT) or a fluoroscopic machine has been used to measure subtalar joint motion [15,16,17]. These techniques may have better reliability and validity than manual examination. However, measuring subtalar joint motion with 4-D CT or a fluoroscopic machine every time in most foot and ankle patients is difficult. The AMA and McBride methods are commonly used to measure subtalar joint ROM in outpatient settings. However, both methods have significant measurement errors. Considering the direction and shape of the anatomical joint surface of the subtalar joint, the McBride method is more consistent than the AMA method in terms of the motion surface of the joint. However, the McBride method does not address the rotation of the lower extremities and does not standardize the degree of the plantarflexion of the ankle joint. To address these points, we used a 3 cm heel pad to standardize the degree of the plantarflexion of the ankle joint.

In the first step in the normal volunteer group, the intraobserver reproducibility of the AMA method was better than that of the modified McBride method. The intraobserver reproducibility of examiner I (foot–ankle specialist) was better than that of examiner II (well trained clinical nurse). The interobserver reliability was also better in the AMA method than in the modified McBride method.

Therefore, we confirmed that the AMA method is more reproducible than the modified McBride method and that a small error existed between the observers. A specialized or expert examiner enhances the intraobserver reproducibility. In the second step in the patient group, the interobserver reliability was evaluated again. Both observers in the patient group were selected as foot–ankle specialists to eliminate the error associated with observer selection.

Interobserver reliability was excellent in the AMA and modified McBride methods. Although confirming the intraobserver reproducibility in the patient group was limited, both methods showed excellent interobserver reliability under similar expertise. In addition to reliability, validity is important, and it indicates the accuracy of the measurement of the characteristics of the objects. It reflects the appropriateness of the measured method.

Two methods were used in this study to verify validity. First, the mean measured values were compared among the three patient groups. A significant decrease (*p* < 0.05) was observed in the AN group (with ankle pathology) and the ST group (with subtalar pathology) compared with that in the NL group (without ankle and subtalar pathology). The AMA method showed no significant difference between the AN and ST groups (*p* > 0.05). However, the modified McBride method showed a statistically significant reduction in the ST group compared with that in the AN group (*p* < 0.05). This finding suggested that the values in the modified McBride method showed a more significant reduction in patients with subtalar joint pathology than in patients with ankle joint pathology. The values in the AMA method also revealed no significant difference between the patients with subtalar and ankle joint pathology. These findings could indicate that the modified McBride method was better than the AMA method in reflecting subtalar joint pathology, including stiffness. The normal ROM of the modified McBride method was wider than that of the AMA method. Therefore, the modified McBride method reflected the subtalar pathology better in a wider range of normal subtalar joint motion. Second, the results of the ROC curve revealed that the modified McBride method showed excellent results. It was better than the AMA method in terms of the AUC and 95% confidence interval.

For example, in a patient treated with subtalar arthrodesis for subtalar arthritis (Figure 4), the results were obtained by two methods, the inversion angle measured by AMA method was 10°, the eversion angle by AMA method was 5°, inversion angle by modified McBride method was 12° and eversion angle by modified McBride method was 6°. Even if the subtalar joints are completely fused, the measured values of AMA method were maintained partially. In another patient undergoing ankle arthrodesis (Figure 5), the inversion angle measured by the AMA method was 10°, and the eversion angle was 6°. A comparison of the two patients showed no difference in the ROM between the patients with ankle fusion and subtalar fusion by the AMA method. Conversely, the modified McBride method for the same patients with subtalar arthrodesis revealed an inversion angle of 12° and an eversion angle of 4°. By contrast, in the patient with ankle fusion, the inversion angle was 20°, and the eversion angle was 11°. Significant differences in the ROM were observed between the two patients by the modified McBride method. Therefore, the AMA method does not accurately reflect pure subtalar joint pathology because of concomitant ankle joint motion. In patients with subtalar joint stiffness, using the modified McBride method led to a significantly smaller ROM than the AMA method.

The AMA method had an error caused by the soft tissue motion of the heel pad and concomitant ankle joint motion. In the modified McBride method in a sitting position, the measurement error could be developed by the motion of adjacent joints such as the Chopart joint and Lisfranc joint.

This study has some limitations. First, we did not consider the differences related to gender and age in the patient group. Second, the observers in the normal volunteer group and the patient group were different. Third, we could not reflect the simultaneous motion of adjacent joints, such as the Chopart joint and Lisfranc joint. Lastly, subtalar joint motion was not evaluated radiologically. Since subtalar joint motion is three-dimensional, it cannot be easily determined with a simple radiologic measurement. As such, further studies should be performed to address these limitations.

## 5. Conclusions

In the normal group, the AMA and modified McBride methods showed fair-to-good intraobserver reproducibility and interobserver reliability. However, the AMA method was relatively more reliable than the modified McBride method with a smaller measurement error. In the patients with subtalar joint stiffness, the modified McBride method had a relatively better validity than the AMA method and was more appropriate for reflecting subtalar joint stiffness.

## Figures and Tables

**Figure 1 jcm-14-01887-f001:**
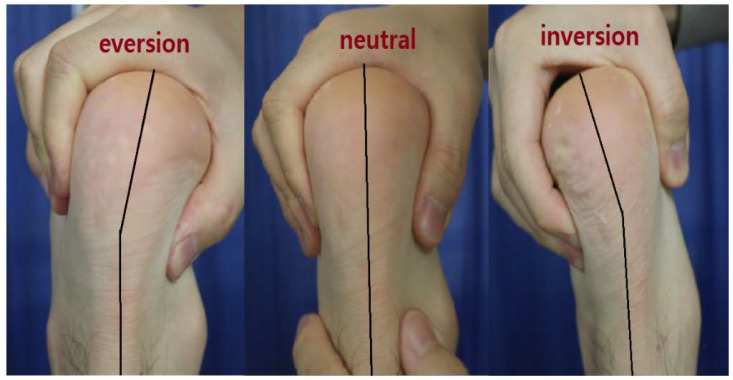
AMA method.

**Figure 2 jcm-14-01887-f002:**
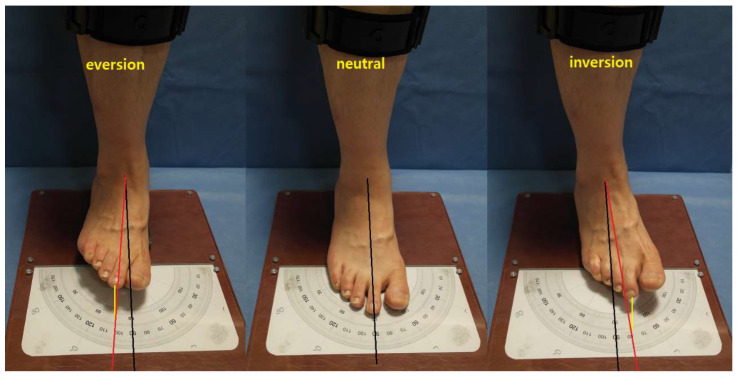
Modified McBride method.

**Figure 3 jcm-14-01887-f003:**
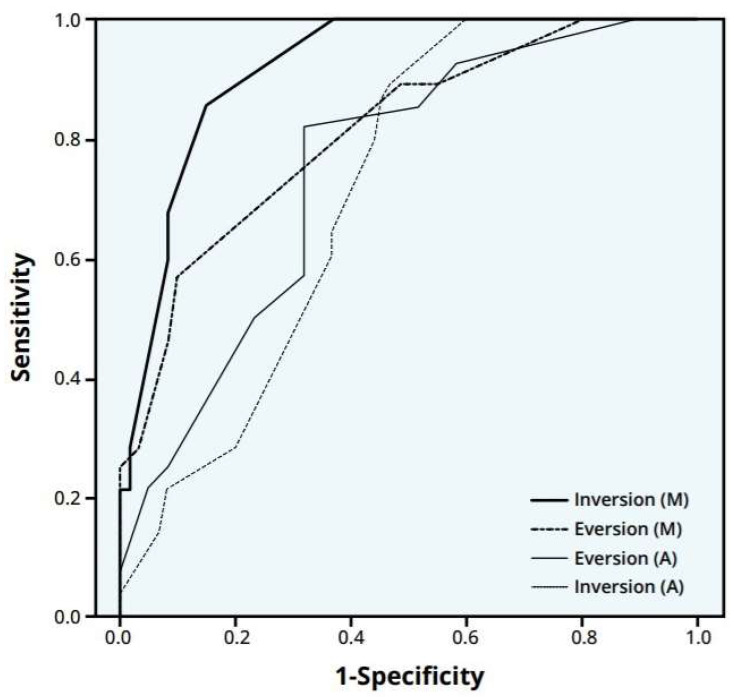
A receiver operating characteristic (ROC) curve. A plot of the true positive rate against the false positive rate for the different possible cutpoints of the two methods. AUC; area under the curve (AUC and 95% CI). AUC 91.9% (95% CI: 86.4–97.4%) for inversion (M)—bold solid line; 81.3% (95% CI: 71.8–90.9%) for eversion (M)—bold dotted line; 74.9% (95% CI: 64.4–85.3%) for eversion (A)—thin solid line; and AUC 71.5% (95% CI: 61.0–81.9%) for inversion (A)—thin dotted line. M: modified McBride method; A: AMA method.

**Figure 4 jcm-14-01887-f004:**
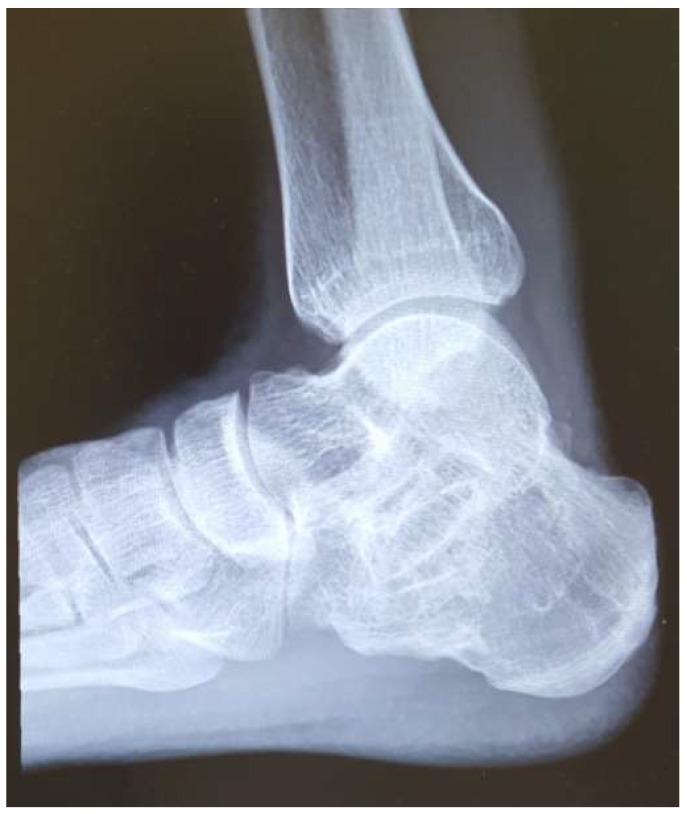
A 40-year-old patient with a fused subtalar joint.

**Figure 5 jcm-14-01887-f005:**
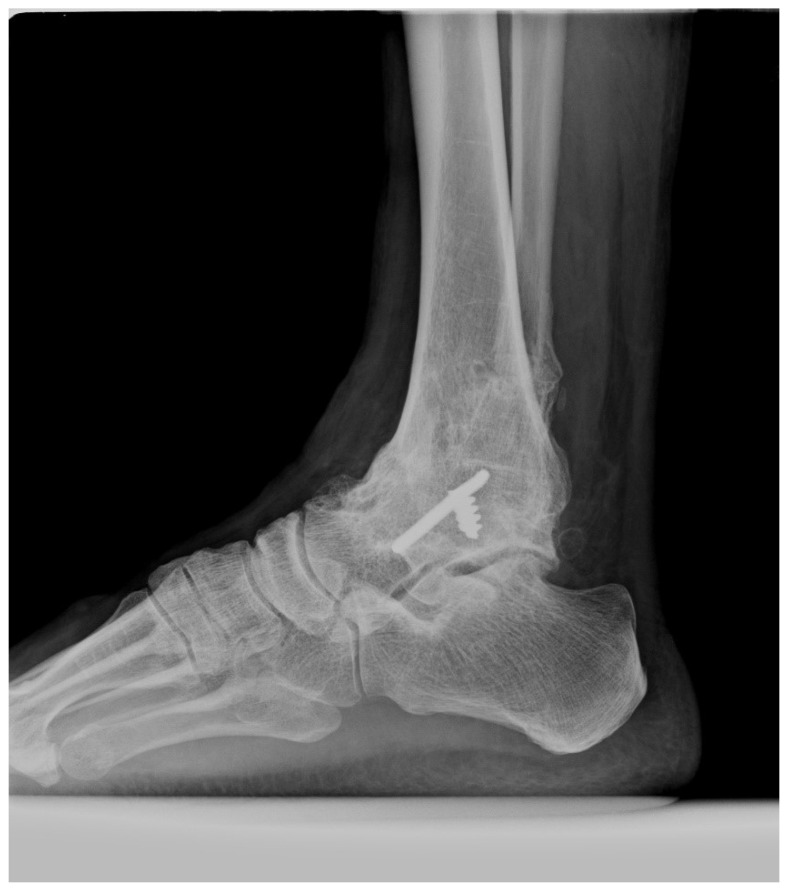
A 57-year-old patient with a fused ankle joint.

**Table 1 jcm-14-01887-t001:** Comparison of demographic data among three patient group.

	NL Group (*n* = 30)	AN Group (*n* = 30)	ST Group (*n* = 30)
Age (yr)	62.5 (30–75)	65.8 (52–78)	61.3 (32–79)
Gender (male–female)	17:13	18:12	17:13
Diagnosis	Normal ankle–subtalar joint	Isolated fused ankle: 21 Advanced ankle OA: 9	Isolated fused subtalar joint: 5 Advanced subtalar OA: 25

**Table 2 jcm-14-01887-t002:** Intraobserver reproducibility in normal group. ICC value from two measurements.

	AMA Method		Modified McBride Method	
	Inversion	Eversion	Inversion	Eversion
Examiner I	0.898	0.746	0.814	0.674
Examiner II	0.857	0.665	0.719	0.568

Examiner I—foot–ankle specialist; examiner II—well-trained clinical nurse.

**Table 3 jcm-14-01887-t003:** Interobserver reliability in normal group. ICC value obtained by two examiners.

	AMA Method		Modified McBride Method	
	Inversion	Eversion	Inversion	Eversion
ICC value	0.905	0.810	0.759	0.750

**Table 4 jcm-14-01887-t004:** Mean ROM in patient groups.

	AMA Method		Modified McBride Method	
	Inversion	Eversion	Inversion	Eversion
NL Group	18.4° (15°~21°)	9.9° (6°~13°)	29.2° (25°~33°)	11.9° (8°~14°)
AN Group	9.7° (6°~12°)	6.4° (3°~9°)	20.3° (14°~28°)	10.9° (6°~13°)
ST Group	10.1° (5°~13°)	5.7° (2°~9°)	15.0° (11°~20°)	8.4° (3°~11°)

NL group: normal ankle and subtalar joint; AN group: ankle joint pathology; ST group: subtalar joint pathology.

**Table 5 jcm-14-01887-t005:** Interobserver reliability in patient group. ICC value obtained by two examiners.

	AMA Method		Modified McBride Method	
	Inversion	Eversion	Inversion	Eversion
ICC value	0.894	0.846	0.921	0.767

**Table 6 jcm-14-01887-t006:** Comparison of validity among 3 groups.

	AMA Method		Modified McBride Method	
	Inversion	Eversion	Inversion	Eversion
NL Grp vs. AN Grp	<0.001	<0.001	<0.001	0.024
NL Grp vs. ST Grp	<0.001	<0.001	<0.001	<0.001
AN Grp vs. ST Grp	0.499	0.356	<0.001	<0.001

NL group: normal ankle and subtalar joint; AN group: ankle joint pathology; ST group: subtalar joint pathology.

## Data Availability

The raw data supporting the conclusions of this article will be made available by the authors on request.

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
