# Peer review of "Which Measuring Method Is Better for Reflecting Subtalar Joint Stiffness?"

_jcm, 2025, doi:10.3390/jcm14061887_

Round 1
Reviewer 1 Report
Comments and Suggestions for Authors
This study was very interesting, and I gained a lot of insights from reading it. I have a few comments for improvement that I hope will be helpful.
Detailed data regarding gender and age distribution are lacking. Since age and gender may influence the measurement results, especially in the patient group, I recommend presenting this information in a table for clarity.
Lines 236–241 discuss why the AMA method may be less suitable for the ST group. However, this section lacks supporting references, which weakens the argument. Including relevant literature on the anatomical and functional aspects of the ankle joint and adjacent joints would strengthen the discussion and improve its overall persuasiveness.
Additionally, it would be beneficial for the authors to elaborate on how the findings of this study could be applied in clinical practice. In particular, providing specific examples of how these methods could be used as simple evaluation tools in outpatient settings would enhance the clinical relevance of the study.
Author Response
Detailed data regarding gender and age distribution are lacking. Since age and gender may influence the measurement results, especially in the patient group, I recommend presenting this information in a table for clarity.
: Thank you for your good suggestion. I summarized the patient information using Table as you suggested.
Lines 236–241 discuss why the AMA method may be less suitable for the ST group. However, this section lacks supporting references, which weakens the argument. Including relevant literature on the anatomical and functional aspects of the ankle joint and adjacent joints would strengthen the discussion and improve its overall persuasiveness.
: To evaluate the validity of these two methods, we compared the AUC values ​​in the ROC curve and the measured average values.
Reviewer 2 Report
Comments and Suggestions for Authors
General characteristics and evaluation of the reviewed article:
The article addresses the important issue of methods for measuring ankle-ankle joint stiffness, comparing two methods: AMA and the modified McBride method. The study includes an assessment of the reliability (ICC) and diagnostic value of the methods using the ROC curve, which is its strong point. The results indicate that the modified McBride method better reflects subacromial joint stiffness compared to the AMA method.
Despite the valuable conclusions, the article has several weaknesses that need improvement to increase the clarity and scientific quality of the paper. Below are my detailed notes and comments on the paper.
Minor comments:
The introduction is to short, it could benefit from further expansion.
Expanding the discussion of osteoarthritis in the introduction could significantly enhance the introduction by highlighting the importance and relevance of this condition. The prevalence of osteoarthritis is influenced by various factors, including occupational activities, sports participation, musculoskeletal injuries, obesity, and gender. Incorporating detailed information about these factors, supported by relevant literature, would provide a robust foundation for the topic. The following references are recommended for inclusion in this section:
https://doi.org/10.3390/healthcare12161648
DOI: 10.1056/NEJMcp1903768
The authors indicate that the study aims to determine which method better reflects the stiffness of the popliteal joint, but there is no clearly formulated research hypothesis. but it is important to clearly state the research hypothesis already in the introduction of the article, such as:
“The research hypothesis is that the modified McBride method reflects the stiffness of the subtalar joint better than the AMA method, which will be verified by ICC and AUC analysis.”
Although there are references to previous work in the article, I feel that after reading the entire paper in detail, some key issues, such as the impact of anatomical individual differences or possible measurement errors, are not sufficiently discussed. It would be worthwhile to add a more detailed comparative analysis of the results in the context of other studies (e.g., those using 3D imaging, radiological methods or those that allow assessment of the joint in motion like vibroarthrography). The authors will find more detailed information in the papers: Multi-Scale Analysis of Knee Joint Acoustic Signals for Cartilage Degeneration Assessment; Application of Recurrence Quantification Analysis in the detection of osteoarthritis of the knee with the use of vibroarthrography; Application of EEMD-DFA algorithms and ann classification for detection of knee osteoarthritis using vibroarthrography; Possible reasons for the differences between the methods studied and the limitations of their use in a clinical setting should be considered.
The authors mention some limitations, such as the failure to account for gender and age-related differences, but do not analyze their potential impact on the results. Possible sampling biases should be discussed in more detail, such as whether the young age of participants in the control group may have influenced better repeatability of measurements. It is worth highlighting the extent to which the methods used can be extrapolated to other patient groups. I recommend expanding the discussion and description of limitations in this regard.
There are language errors and awkward wording in the text (e.g., “This finding suggested that the values in the modified McBride method showed a more significant decline in patients with subtalar joint pathology than in patients with ankle joint pathology.”). Language correction is recommended, especially for clarity and conciseness of statements. Some sentences should be corrected to avoid repetition and to make the narrative more coherent.
It is not entirely clear whether the patients in the pathology groups had homogeneous characteristics regarding disease progression. It would be useful to provide more precise criteria for selecting patients (e.g., the severity of degenerative changes). A good supplement would be to provide information about possible differences in disease duration and their possible impact on the measurements.
Although the article includes tabulated results, it lacks clear graphs that could better illustrate the relationships between the variables studied. It is recommended to add graphs showing the ICC and AUC results for both methods. It is also possible to present the data in the form of a bar chart with a comparison of the average ROM values for the different study groups.
Comments on the Quality of English Language
There are language errors and awkward wording in the text (e.g., “This finding suggested that the values in the modified McBride method showed a more significant decline in patients with subtalar joint pathology than in patients with ankle joint pathology.”). Language correction is recommended, especially for clarity and conciseness of statements. Some sentences should be corrected to avoid repetition and to make the narrative more coherent.
Author Response
The introduction is to short, it could benefit from further expansion.
: Thanks for your suggestions. Based on the recommended paper, we have added content to the introduction section, mentioning OA.
The authors indicate that the study aims to determine which method better reflects the stiffness of the popliteal joint, but there is no clearly formulated research hypothesis. but it is important to clearly state the research hypothesis already in the introduction of the article, such as:
“The research hypothesis is that the modified McBride method reflects the stiffness of the subtalar joint better than the AMA method, which will be verified by ICC and AUC analysis.”
: Thanks for your suggestion. I modified the hypothesis according to your advice.
Although there are references to previous work in the article, I feel that after reading the entire paper in detail, some key issues, such as the impact of anatomical individual differences or possible measurement errors, are not sufficiently discussed. It would be worthwhile to add a more detailed comparative analysis of the results in the context of other studies (e.g., those using 3D imaging, radiological methods or those that allow assessment of the joint in motion like vibroarthrography). The authors will find more detailed information in the papers: Multi-Scale Analysis of Knee Joint Acoustic Signals for Cartilage Degeneration Assessment; Application of Recurrence Quantification Analysis in the detection of osteoarthritis of the knee with the use of vibroarthrography; Application of EEMD-DFA algorithms and ann classification for detection of knee osteoarthritis using vibroarthrography; Possible reasons for the differences between the methods studied and the limitations of their use in a clinical setting should be considered.
: The original purpose of this paper was not to accurately measure range of motion.
The question is “which of the two methods reflects joint stiffness better? “
Therefore, we believe that the results of other papers using 3D imaging or other radiologic methods are not the center
of controversy in our paper.
The authors mention some limitations, such as the failure to account for gender and age-related differences, but do not analyze their potential impact on the results. Possible sampling biases should be discussed in more detail, such as whether the young age of participants in the control group may have influenced better repeatability of measurements. It is worth highlighting the extent to which the methods used can be extrapolated to other patient groups. I recommend expanding the discussion and description of limitations in this regard.
: Demographic data of patient group was added and summarized in Table 1. However, the differences between these factors were not taken into account statistically. This point was mentioned as a limitation in the discussion section.
There are language errors and awkward wording in the text (e.g., “This finding suggested that the values in the modified McBride method showed a more significant decline in patients with subtalar joint pathology than in patients with ankle joint pathology.”). Language correction is recommended, especially for clarity and conciseness of statements. Some sentences should be corrected to avoid repetition and to make the narrative more coherent.
: Thanks for the advice. The word decline in this sentence is unified with the term in other sentences as reduction. Contrary to your concerns, this paper has already been proofread in English before submission.
It is not entirely clear whether the patients in the pathology groups had homogeneous characteristics regarding disease progression. It would be useful to provide more precise criteria for selecting patients (e.g., the severity of degenerative changes). A good supplement would be to provide information about possible differences in disease duration and their possible impact on the measurements.
: Patients who underwent ankle joint and subtalar joint fusion are considered to have homogeneous characteristics. Other patients with end-stage ankle OA or post-traumatic subtalar OA may have heterogeneous characteristics but have little range of motion in the relevant joint.
Although the article includes tabulated results, it lacks clear graphs that could better illustrate the relationships between the variables studied. It is recommended to add graphs showing the ICC and AUC results for both methods. It is also possible to present the data in the form of a bar chart with a comparison of the average ROM values for the different study groups.
: Following your advice, I added and inserted the AUC value into the ROC curve. I will change with an new figure file as soon as the figure work is finished.